# Missed Diagnosis of Major Depressive Disorder with Catatonia Features

**DOI:** 10.3390/brainsci9020031

**Published:** 2019-02-02

**Authors:** Harry Jhawer, Meesha Sidhu, Rikinkumar S. Patel

**Affiliations:** 1School of Medicine, Avalon University, Girard, OH 44420, USA; 2Department of Psychiatry, Griffin Memorial Hospital, Norman, OK 73071, USA; meesha.sidhu@gmail.com (M.S.); dr.rknpatel@gmail.com (R.S.P.)

**Keywords:** catatonia, MDD, depression, BZD, benzodiazepine, anxiety, lorazepam, clinical improvement

## Abstract

Catatonia is often a presentation of extreme anxiety and depression. Missing the diagnosis of catatonia would lead to improper treatment, which could be life-threatening. A thorough physical and psychiatric assessment is required for detecting the catatonic symptoms, especially, mutism and negativism in patients with depression. We discuss the case of a 58-year-old female that was incorrectly diagnosed and treated for major depressive disorder (MDD). The patient was then correctly diagnosed with MDD with catatonic features and improved once benzodiazepine (BZD) was started. The preferred BZD was lorazepam, with a success rate of complete remission of up to 80% in adults. Treatment was started with lorazepam 1–2 mg and improvement was seen within the first ten minutes. We believed the addition of BZD in a psychotropic regimen could improve both catatonia and depression, and should be continued for 3–6 months to prevent relapses and recurrences.

## 1. Introduction

The number of catatonic patients among acutely ill psychiatric inpatients varies from 7.6 to 38% [1]. A higher proportion of catatonic patients have comorbid bipolar disorder (43%) [2] and schizophrenia (30%) [1]. Currently, there are three significant subtypes of catatonia, namely, retarded, excited, and malignant catatonia [3]. The presentation of catatonia falls between retarded and excited subtypes, and rarely presents as a hallmark picture of either.

Catatonia is often a presentation of extreme anxiety [4]. Missing the diagnosis of catatonia would lead to improper treatment, which could be life-threatening as it may lead to arrhythmia or hyperthermia [5]. Furthermore, treating catatonia with an antipsychotic increases the patients’ risk of developing neuroleptic malignant syndrome [6]. With an improper diagnosis, they would not be put on the appropriate preventive care such as deep vein thrombosis, pulmonary embolism, contractures, and pressure ulcers [7].

Here, we present a case of depression with catatonic symptoms diagnosed after meticulous observation and psychiatric evaluation. The change in the diagnosis led to the addition of benzodiazepine (BZD) in inpatient management, which ultimately led to drastic improvements in the patient outcomes.

## 2. Case Presentation

This case report conformed to the Declaration of Helsinki. Case that is discussed herein was only stated after getting a verbal consent from the patient and approval of the State Hospital. Ms. L, a 58-year-old Caucasian female, was transferred to us from another inpatient psychiatric facility for further management. She had a long-standing history of depression and anxiety disorders. As per their notes, she met the Diagnostic and Statistical Manual of Mental Disorders, 5th edition (DSM-V) diagnostic criteria of major depressive disorder (MDD), recurrent episode. They started the patient on mirtazapine 15 mg twice daily, fluoxetine 20 mg in the morning for depression, and buspirone 15 mg twice daily for anxiety.

During our intake, the patient complained of anxiety resulting in a decrease in her function of day-to-day living. The patient stated she had limited intake of fluids for the past three weeks. She described her mood as “I do not have moods; I lost the ability to cry or do anything.” Patient’s insight into her underlying illness and judgment was limited.

She was continued on her medical regiment and started on a soft diet and meal replacement shakes due to her decreased appetite. The next day, the patient was interviewed in her room as she refused to get out of her bed. She had a minimal verbal response and also showed negativism by refusing to participate with the treatment team. The patient diagnosis was changed to MDD with catatonic features. She was immediately started on lorazepam 1 mg twice daily, while continuing her other psychiatric medications. Within a few hours, the patient was seen outside her room and communicating with staff. The patient also stated that her appetite was returning.

The patient continued to improve as her inpatient management continued, eventually denying anxiety and depression. She remained compliant with her psychotropic regimen. On day 6, the patient was taken off from suicide and self-harm precautions as she denied suicidal ideation. The patient was discharged with the antidepressant regimen and was continued on lorazepam for catatonic symptoms.

## 3. Discussion

The findings of this case show the importance of a thorough physical and psychiatric assessment for detecting catatonic symptoms in patients with MDD. Our patient’s catatonic symptoms were either missed or not recognized by the initial treatment team. During her admission at our facility, we identified the patient’s reluctance to get off the bed, partial mutism, and negativism to fulfill criterion A of the DSM-V’s criteria for MDD subtype catatonia. MDD caused our patient’s catatonia with severe anxiety, and no associated delirium, which fulfilled criteria B, C, and D. Also, the patient was not functional due to the severity of the symptoms (meeting criteria E). There were two key differentials for this patient—serotonin syndrome and elective mutism. The patient did not meet the criteria for Hunter serotonin toxicity decision rules, thus ruling out serotonin syndrome. She also lacked a history of personality disorder that was often accompanied in patients of elective mutism.

A prospective study by Worku and Fekadu (2015) stated that catatonia involved the reduction or inhibition of the GABA receptors that connected the basal ganglia with the cortex and thalamus in the right orbitofrontal lobe. Of note, among catatonic patients, only the right orbitofrontal activity was reduced; the left orbitofrontal lobe activity remained unchanged [4]. Patients with anxiety also showed a reduction in activity in the orbitofrontal area [8]. We believe the success of BZD in our patients could be due to the alleviation of anxiety. It could be hypothesized that lorazepam decreased our patient’s anxiety with improvement in catatonic symptoms, which indirectly led the patient to function at full potential.

For patients that are being treated for MDD subtype catatonia, the preferred BZD is lorazepam, with a success rate of complete remission of up to 80% in adults and 65% in children [2]. As per the guidelines, treatment is started with 1–2 mg of lorazepam every four to twelve hours [2]. Improvement in patients can sometimes be seen within the first ten minutes after initiating lorazepam [9]. Overall remission with BZD is achieved within four to ten days, although the patient is managed for a total 3–6 months with a slow tapering. And, for the patients who do not respond to BZD, Electroconvulsive therapy (ECT) or a combination of BZD and ECT was recommended [10].

It is essential to obtain a detailed psychiatric history of patients with severe MDD to observe for certain catatonic features (including mutism and negativism), which can be often missed. Treatment for MDD subtype catatonia differs from MDD. For MDD subtype catatonia, adding a BZD has shown to improve both catatonia and depression while also preventing relapses and recurrences.

## 4. Conclusions

Timely intervention of catatonia becomes vital to prevent any long term morbidities. In patients with MDD subtype catatonia, the symptoms of catatonia may get confused or over looked as symptoms of MDD. Clinicians need to maintain a high level of suspicion. BDZ have a proven record of success in treating catatonia.

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
