# Peer review of "Missed Diagnosis of Major Depressive Disorder with Catatonia Features"

_brainsci, 2019, doi:10.3390/brainsci9020031_

Round 1
Reviewer 1 Report
This is a case-report on a 58 female with history of depression, anxiety presenting to an inpt psychiatric (?) unit with anxiety. During admission, she developed catatonic symptoms.
I am not convinced that her catatonic symptoms were due to her mood/anxiety disorder, and not, for example, changes in medication (starting mirtazepine, fluoxetine, buspirone). Additionally, other differentials should be discussed (and ruled out if appropriate), such as serotonin syndrome.
Throughout the manuscript, sentence structure should be edited and typos fixed (ex. p. 1 lines 22, 32, 39-41; p. 2 lines 72-73, 78-79).
The concluding paragraph should be toned down. BZD should not be used to treat depression. It is also dangerous to make the statement that all patients should remain on BZD to prevent relapses/recurrences (risk of falls, dependence, interactions, etc.).
Author Response
I am not convinced that her catatonic symptoms were due to her mood/anxiety disorder, and not, for example, changes in medication (starting mirtazepine, fluoxetine, buspirone). Additionally, other differentials should be discussed (and ruled out if appropriate), such as serotonin syndrome.
I appreciate your feedback.
I added two differentials and further support to help the argument that her mood/anxiety disorder lead to her catatonic symptoms.
The concluding paragraph should be toned down. BZD should not be used to treat depression. It is also dangerous to make the statement that all patients should remain on BZD to prevent relapses/recurrences (risk of falls, dependence, interactions, etc.).
I have specifically stated that the treatment plan refers to MDD with catatonic features, not just MDD.

Reviewer 2 Report
Having read the manuscript entitled: “Why lorazepam helped a patient with major depressive disorder?” I have to admit that the paper is generally well-written. Both the topic as well as the findings are interesting. The case is presented clearly and logically. I only have several minor comments:
1. I suggest to change the title of the manuscript, since the Discussion section did not answer the question posed in the title. Though the Authors give some explanation but it is only an assumption. The described case report just demonstrates that lorazepam helped a patient with major depression, but it does not answer the question “why”.
2. When reading an Abstract one can assume that the manuscript is a review instead of a case report. The Authors did not mention that they present a case of depression with catatonic symptoms that was successfully managed by lorazepam.
3. The Introduction section lacks a short paragraph on association between catatonia and major depressive disorder.
4. In the Discussion section the Authors could mention the study by Hung and Huang
Lorazepam and diazepam rapidly relieve catatonic features in major depression. Clin Neuropharmacol. 2006 May-Jun;29(3):144-7.
Author Response
Thank you for your feedback.
I suggest to change the title of the manuscript, since the Discussion section did not answer the question posed in the title. Though the Authors give some explanation but it is only an assumption. The described case report just demonstrates that lorazepam helped a patient with major depression, but it does not answer the question “why”.
I have revised the title.
2. When reading an Abstract one can assume that the manuscript is a review instead of a case report. The Authors did not mention that they present a case of depression with catatonic symptoms that was successfully managed by lorazepam.
I have added more details clarifying that this is a case study.
3. The Introduction section lacks a short paragraph on association between catatonia and major depressive disorder.
The association between catatonia and MDD is not well understood, and not much is offered in current literature.
I have added in the discussion some association between MDD and and catatonia.

Round 2
Reviewer 1 Report
Please state if the pt remained on the antidepressants throughout her hospitalization. Additionally, a few typos should be corrected.
Author Response
Thank you for the feedback. I have added a sentences that clarifies that the patient's other medications were continued.
She was continued on her medical regiment and started on a soft diet, and meal replacement shakes due to her decreased appetite. The next day, the patient was interviewed in her room as she refused to get out of her bed. She had a minimal verbal response and also showed negativism by refusing to participate with the treatment team. The patient diagnosis was changed to MDD with catatonic features. She was immediately started on lorazepam 1mg twice daily while continuing her other psychiatric medications. Within a few hours, the patient was seen outside her room and communicating with staff. The patient also stated that her appetite was returning.
